# Vicarious Traumatization Questionnaire: Psychometric Properties Using Rasch Model and Structural Equation Modeling

**DOI:** 10.3390/ijerph18094949

**Published:** 2021-05-06

**Authors:** Mohd Noor Norhayati, Samah Ali Mohsen Mofreh, Yacob Mohd Azman

**Affiliations:** 1Department of Family Medicine, School of Medical Sciences, Health Campus, Universiti Sains Malaysia, Kubang Kerian 16150, Malaysia; hayatikk@usm.my; 2School of Educational Studies, Universiti Sains Malaysia, Gelugor 11800, Malaysia; 3Hospital Raja Perempuan Zainab II, Kota Bharu 15586, Malaysia; drmohdazman@moh.gov.my

**Keywords:** vicarious traumatization, healthcare providers, psychometric properties, Rasch model, structural equation modeling

## Abstract

Frontline healthcare providers are exposed to indirect trauma through dealing with traumatized patients. This puts them at risk of vicarious traumatization. In response to the COVID-19 pandemic, this study seeks to establish the psychometric properties of the Malay version of the Vicarious Traumatization Questionnaire among healthcare providers. A cross-sectional study was conducted. The translated Malay version of the Vicarious Traumatization Questionnaire was completed by 352 healthcare providers in Kelantan, Malaysia. The data was entered using IBM SPSS Statistics version 26.0 (SPSS Inc., Chicago, IL, USA, 2019), and descriptive analysis was performed. The psychometric properties of the scale were assessed in two phases. The Rasch model to assess the validity and reliability was performed using Winsteps version 3.72.3. The confirmatory factor analysis using the structural equation modeling was performed using AMOS version 23.0. The Rasch analysis showed that the 38 items, in two constructs, had high item reliability and item separation at 0.97 and item separation at 5.36, respectively, while good person reliability and person separation were at 0.95 and 4.58, respectively. The correlations of all persons and items are greater than 0.20. There are no misfitting or overfitting items in the outfit MNSQ. There are four items that are challenging in answering the scale. The final model of the confirmatory factor analysis shows two constructs with 38 items demonstrating acceptable factor loadings, domain to domain correlation, and best fit (Chi-squared/degree of freedom = 4.73; Tucker-Lewis index = 0.94; comparative fit index = 0.94; and root mean square error of approximation = 0.10). Composite reliability and average variance extracted of the domains were higher than 0.7 and 0.5, respectively. The Vicarious Traumatization Questionnaire tested among healthcare providers has been shown to valid and reliable to assess vicarious traumatization.

## 1. Introduction

The escalating pandemic of COVID-19 strides a vast impact physically and emotionally on frontline healthcare providers. In response to the COVID-19 pandemic, frontline healthcare providers are highly vulnerable to experiencing physical, mental, and emotional exhaustion. Besides the shortage of staff, frontline healthcare providers face the possibility of being exposed to and infected with COVID-19 [1].

Frontline healthcare providers are often exposed to indirect trauma by working with traumatized patients. This puts them at risk of developing vicarious traumatization, resulting in disruptions of normal cognitive schemas, with emerging symptoms of negative cognition, mood, low concentration, flashbacks, bad dreams, and intrusive thoughts or memories of the trauma of the patients [2]. Exaggerated with overworked, fatigue, and lack of support, vicarious traumatization can eventually contribute to the feeling of being burdened, overwhelmed, and hopelessness in the face of great need and suffering of the patients and pandemic situations [3,4]. This kind of mental health problem not only affects the frontline health providers’ attention, understanding, and decision-making ability, which might hinder the fight against COVID-19; but could also have a lasting effect on their overall well-being [5].

Vicarious traumatization refers to harmful changes to oneself due to exposure to others’ traumatic events [6]. In this study, the research tool is based on the validated original Chinese version of the vicarious traumatization questionnaire. The original version was developed from qualitative interviews and adapted international trauma-related scales [7]. A tool for assessment of such an impact on healthcare providers is very much needed, to identify those who are at risk so that further intervention can be taken to preserve their general well-being.

This study aimed to establish the psychometric properties of the Malay version of the Vicarious Traumatization Questionnaire from its original Chinese version among healthcare providers in response to the COVID-19 pandemic in Kelantan. Frontline healthcare providers refer to those engaged in providing care for patients with COVID-19 who have been pre-identified in the healthcare facility involved.

## 2. Materials and Methods

### 2.1. Population and Sample

A cross-sectional study was conducted among healthcare providers between May and July 2020 in two hospitals managing persons under investigation and COVID-19 cases in Kelantan, Malaysia. Healthcare providers, including doctors, nurses, and medical assistants, were included. Those diagnosed to have any psychiatric illnesses were excluded. Convenient sampling of healthcare providers was applied. Those who responded to the survey in the WhatsApp application and fulfilled the eligibility criteria were included. The sample size was based on definitive or high stakes at 99% confidence with best to poor targeting sample size of more than 250 samples [8].

### 2.2. Research Tools

In this study, the Vicarious Traumatization Questionnaire underwent a translational process outlined by the Translation and Cultural Adaptation-Principles of Good Practice [9] to ensure the content and face validity. The Vicarious Traumatization Questionnaire consists of 38 items, which are composed of two constructs: physiological (11 items) and psychological (27 items). The psychological construct is subdivided into four subconstructs. Namely, emotional (nine items), behavioral (seven items), cognitive (five items), and life belief (six items) subconstructs [7]. Each question score ranged from 0 (never) to 5 (always). Total raw scores are used. The score ranges from 0 to 190, with higher scores indicating more vicarious traumatization. Cronbach’s alpha for the questionnaire was 0.93, and the values for each dimension ranged from 0.73 to 0.92.

### 2.3. Data Collection

Eligible respondents were invited through an online method. A google form was distributed through the group WhatsApp application. The respondents who agree to participate in the research were requested to respond to a virtual consent form and complete the self-administered questionnaire. Respondents were informed that their participation in research was voluntary, and that it was permissible for them to withdraw. The respondents were not required to sign-in to a Google account to fill in the survey. Respondents who were identified to be at risk of developing significant psychological conditions were referred for counselling and psychological support to the Psychological First Aid team in the hospital.

### 2.4. Statistical Analysis

The data was entered using IBM SPSS Statistics version 26.0 (SPSS Inc., 2019), and descriptive analysis was used to analyze healthcare participants’ sociodemographic characteristics. This study assessed the psychometric properties of the established original Chinese version Vicarious Traumatization Questionnaire, which entails the formulation of a measurement model in two phases.

The first phase was to perform the validity and reliability using the Rasch model analysis with Winsteps version 3.72.3 [10]. An instrument’s validity can be identified in the Rasch model based on item polarity, item and person map, misfit and infit items, item and person separation, dimensionality, and scale calibration [11].

The reliability acceptance requirements in the Rasch model should exceed 0.50 [11,12]. Appropriate separation value should exceed 2 [13]. In Rasch analysis, the reliability of the persons and items was equivalent to Cronbach’s alpha. For the analysis of these construct items, the result expected mean square (MNSQ) infit analysis value should be 0.5 < x <1.5, and the point-measure correlation (PTMEA) value should be +0.2 < x < 1 [11].

The second phase was to perform the confirmatory factor analysis (CFA) using structural equation modeling (SEM) with AMOS version 23.0. Several steps were tested through CFA analysis, which is constructing a path diagram, assessing model identification, evaluating estimates and model fit, interpreting and analyzing the initial model, and the final model. The goodness of fit was measured using chi-square for the null hypothesis significance test [14,15]. Convergent validity, construct validity, and discriminative validity was evaluated for the validity of the measurement model.

The factor loadings, composite reliability scores (CRs), and average variance extracted scores (AVEs) were used to address the convergent validity and discriminant validity. The Composite Reliability (CR) for the CFA analysis is intended to determine the consistency of construct validity indicator. The CR was estimated by using the equation (CR = Square of standardized loading/Square of total standardized loading + measurement errors). The cut-off value for CR of more than 0.7, indicating high internal constancy. The convergent validity was verified through Average Variance Extracted (AVE), and the AVE should be greater or equal to 0.5. The AVE was calculated for the measurement model by calculating the sum of the variance of constructs and then dividing it by the number of constructs of the Vicarious Traumatization Questionnaire. The discriminative validity of the initial model was reached when the model of measurement correlation between and pair of latent exogenous constructs was less than 0.85. The comparative fit index (CFI), standardized root mean-variance and covariance of the variables, and factor loadings and residuals are parameter estimates for this study. On the latent variables, metrics should have coefficients (factor loadings) of 0.6 or higher [14].

## 3. Results

A total of 352 healthcare providers participated in this study by completing the questionnaire. The mean (standard deviation, SD) of age was 38.2 (6.80) years, and the majority of the participants were females (80.4%). Other sociodemographic characteristics of the participants are presented in Table 1.

### 3.1. Rasch Model Analysis

Rasch analysis was used to test the reliability of 38 items for Vicarious Traumatization Questionnaire. Data analysis of the person and item reliability using the Rasch model were shown (Table 2). The person reliability was very high at a value of 0.95, and the person separation was 4.58, with item reliability at 0.97 and item separation at 5.36, which were acceptable. Analysis of the study showed that the reliability of 352 respondents with 38 items in these constructs was high to measure the Vicarious Traumatization Questionnaire. Thus, the reliability of items for the scale values was reasonably close together, and both were representing a strong acceptable level.

A further investigation into the item misfit statistics was conducted—parameters for the 38 item statistics measured between 0.69 to −1.18 logit. The outfit MNSQ was 1.51 to 0.64 logit, the outfit z-std was 5.1 to −4.20 logit, and the PTMEA was between 0.75 to 0.59 logit. The correlations of all items were positive and greater than 0.20. The positive values show that all persons and all items were moving in parallel to measure the constructs formed. There were no misfitting or overfitting items in the outfit MNSQ. Therefore, the data are deemed acceptable for this study.

The dimensionality analysis result shown in Table 3 describes the direction and dimension of the questionnaire. The raw variance explained by measures was 56.4%, and the unexplained variance in the 1st contrast was 5.2%. Thus, dimensionality data shows that the scale has a satisfactory dimensionality, which was determined by the raw variance explained by measures of more than 40% and unexplained variance in 1st contrast.

Rasch analysis determines the validity of the response probabilities being spread fairly across scales. Table 4 and Figure 1 show a summary of the category structure on a scale gradation and size structure of the intersection. The column arrangement observation (observed count) shows the respondents’ answers given to the ranking scale. The most frequent answer response is category 3 (*n* = 147, 43%). The next grading scale that respondents selected was scale 2 of 1267 (37%). Category 1 had 53 (15%) respondents. The observed averages show the pattern of respondents. A fairly normal pattern is expected with a systematic instrument from negative to positive, signifying that the respondents’ answers are fairly normal. Therefore, the calibration scaling analysis shows good validity with the five scales of this instrument, and all five scales, including 1, 2, 3, 4, and 5, respectively, can be used.

Figure 2 shows the difficulty of items and the ability of persons of the Vicarious Traumatization Questionnaire scale among the respondents. The majority of respondents could not answer all items. This result shows that respondents’ ability was lower to answer difficult questions indicating that the person’s ability to face the vicarious traumatization was high compared to their ability. The results show that the most challenging items in answering the scale were item VT-Q23CO (“I seemed to hear a call for help”), VT-Q15BE (“I do not feel determined to do things”), VT-Q30PH (“I have no appetite”), and VT-Q06BH (“I deliberately avoid some topics and situations related to the disaster”), respectively.

### 3.2. Confirmatory Factor Analysis

The unidimensionality was achieved when measuring items having acceptable factor loading equal to or higher than the value of 0.5 for the respective latent construct (Figure 3). The construct items had good satisfactory factor loadings, hence, signifying unidimensionality for the initial model. The AVE of Vicarious Traumatization Questionnaire constructs was 0.73. The results of AVE indicated that all items in the measurement model were statistically significant. The construct items had good satisfactory factor loadings higher than 0.5. Hence, representing the unidimensionality of the initial model.

Table 5 indicates the index level for acceptable goodness fit for the construct validity of the initial model. The goodness fit indices were acceptable except for CFI, TLI, and IFI, which indicated low goodness fit. Therefore, some modifications were made to improve the initial measurement model.

Modifications were made to improve the initial measurement of the model. Figure 4 and Table 6 depicted a revised measurement model. The indices improved and showed an acceptable goodness fit of the revised measurement model for the Vicarious Traumatization Questionnaire.

Apply the CR equation, the estimates of each of all items of the standardized regression were performed, as shown in Table 7. The CR was 0.95, indicating very high internal constancy for the two constructs of physiological and psychological responses, while the AVE result for the revised model was 0.68 for both constructs. The satisfaction of conditions for all the regression weights, CR, and AVE support the convergent validity of the Malay version of the Vicarious Traumatization Questionnaire.

Table 8 shows the estimated correlations of the two constructs, namely the Vicarious Traumatization Questionnaire’s physiological and psychological constructs. There is a strong significant relationship between physiological and psychological constructs with an *r*-value of 0.83.

## 4. Discussion

The psychometric properties of the Malay version of the Vicarious Traumatization Questionnaire among healthcare providers in response to the COVID-19 pandemic were tested in two stages using the Rasch model and CFA. The Rasch analysis resulted in a similar factor model (the exploratory model) with the theoretically specified model. This indicates that the Malay version of the Vicarious Traumatization Questionnaire consisting of two constructs and 38 items can be used to measure vicarious traumatization. Based on the CFA results, it was concluded that the exploratory model fit with the theoretical model.

The exploratory-theoretical model incorporated the theoretical separation of the Vicarious Traumatization Questionnaire into five factors correlated to one another. The exploratory model also provided a better fit than this exploratory-theoretical model. Based on the CFA results, it can be concluded that while none of the models provides absolute fit, the exploratory model provides the best fit. This conclusion is based on the comparative fit statistics, IFI, TLI and CFI, and the scale reliabilities. Therefore, the CFA results strongly show a positive and significant relationship between physiological response and psychological responses as main components of the questionnaire with strong positive correlations among the scale items and how they correlated significantly to their perspective constructs.

All values of the model-of-fit indices, including the absolute fit indices and comparative indices, are acceptable. The RMSEA values are acceptable for the exploratory and the exploratory-theoretical model and bordering acceptance for the two theoretical models. The results thus provide enough evidence to draw two important conclusions. First, evidence was found both in the Rasch model and in the CFA that the Vicarious Traumatization Questionnaire factors are similar. Rasch analysis results indicate that the translated questionnaire showed good overall fit, item fit, targeting, and internal consistency. Therefore, all items had ordered thresholds, there was no response dependence, items were unidimensional, and there was no evidence of differential item functioning.

The results of the CFA further confirmed this finding. The absolute and the comparative fit statistics clearly show a better fit for the theoretical model without factors and dimensions of Vicarious Traumatization Questionnaire compared to the theoretical model with factors and dimensions of Vicarious Traumatization Questionnaire. Therefore, based on the present study, it is not advisable to include task strategies as a separate scale in the Vicarious Traumatization Questionnaire’s factors and dimensions because this could jeopardize the instrument’s validity.

Assessing the psychometric properties is vital for any instrument to be used as a reliable and valid measurement tool (Mofreh, 2020). Analyzing the psychometric properties of this scale among healthcare providers enables it to be located [16] and modified for use in the local context. This is the first study that assesses the psychometric properties of the Vicarious Traumatization Questionnaire using the Rasch model and structural equation modeling.

## 5. Conclusions

The Malay version of the Vicarious Traumatization Questionnaire tested among healthcare providers is valid and reliable after testing for the person, and item fit statistics and polarity and confirming the construct validity.

## Figures and Tables

**Figure 1 ijerph-18-04949-f001:**
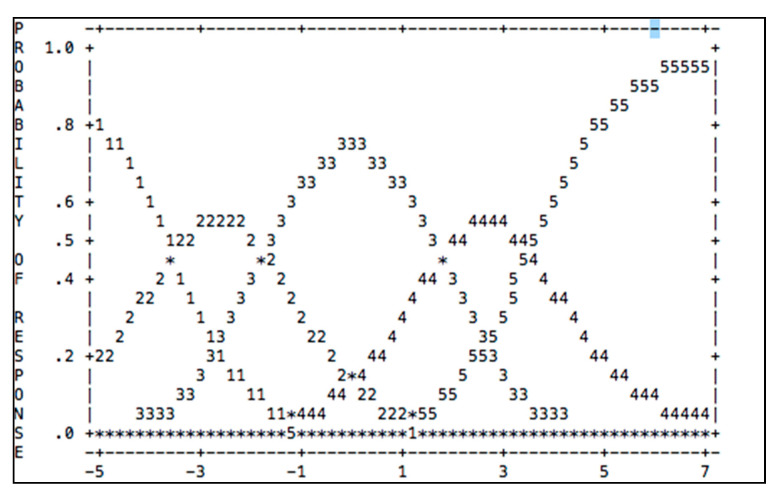
Summary of the category structure on a scale gradation.

**Figure 2 ijerph-18-04949-f002:**
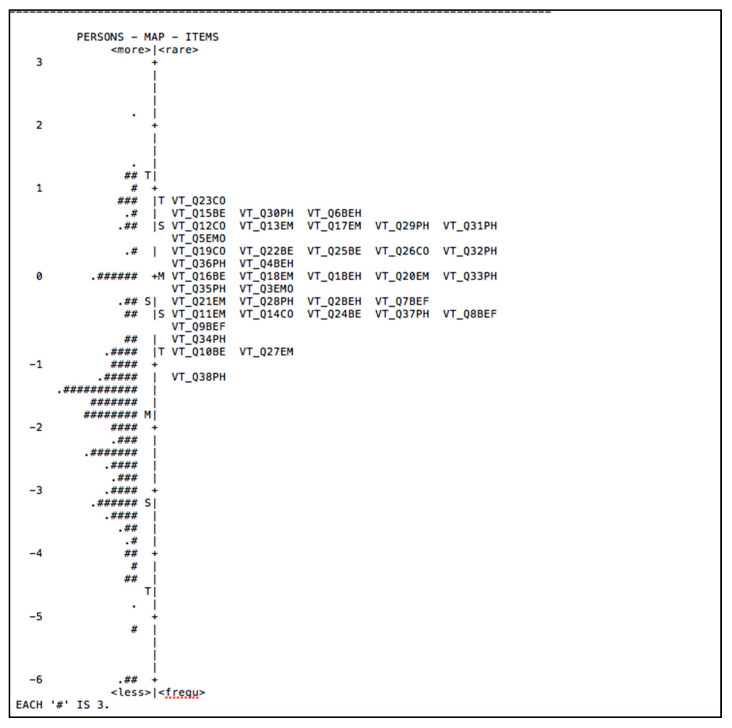
The item-person map of Vicarious Traumatization Questionnaire.

**Figure 3 ijerph-18-04949-f003:**
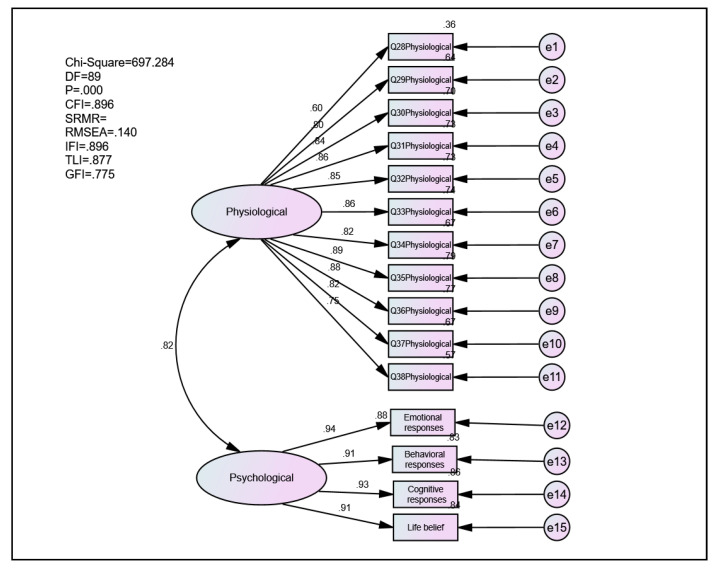
The initial measurement model.

**Figure 4 ijerph-18-04949-f004:**
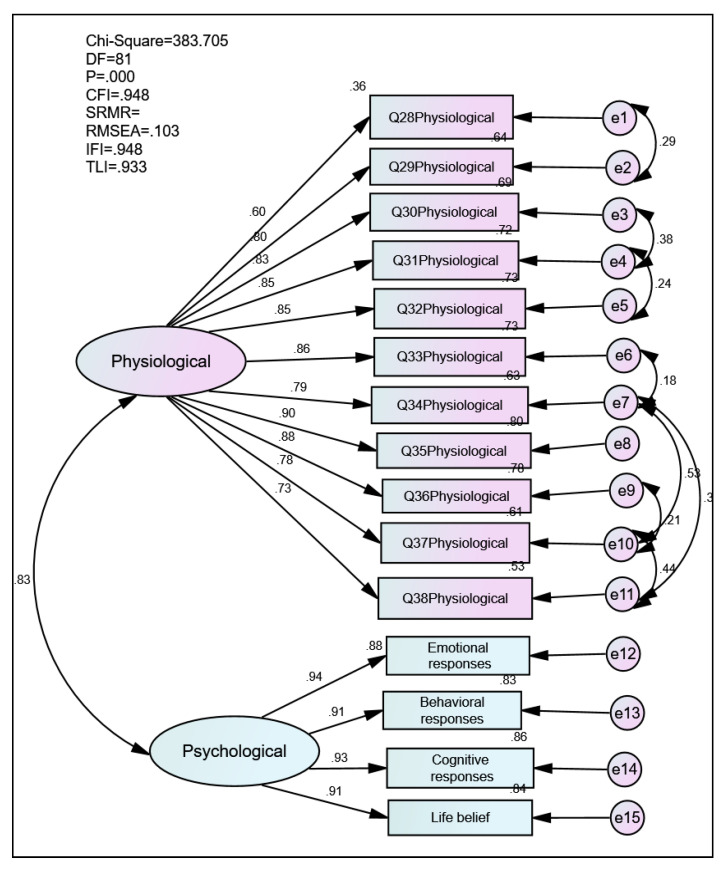
Measurement model.

**Table 1 ijerph-18-04949-t001:** Sociodemographic characteristics of participants (*n* = 352).

Variables	Mean	(SD)	*n*	(%)
Age (years)	38.2	(6.80)		
Household income (MYR)	5382.4	(3039.90)		
Number of children	2.4	(1.58)		
Sex				
Male			69	(19.6)
Female			382	(80.4)
Race				
Malay			347	(98.6)
Non-Malay			5	(1.4)
Education level				
Diploma			303	(86.1)
Bachelor			36	(10.2)
Master			13	(3.7)
Marital status				
Married			305	(86.6)
Unmarried			47	(13.4)
Shift work				
No			45	(12.8)
Yes			307	(87.2)
Administrative work				
No			335	(95.2)
Yes			17	(4.8)

Note: SD = Standard Deviation.

**Table 2 ijerph-18-04949-t002:** Person and item summary statistics.

	Person (*n* = 200)	Item (*n* = 27)
Reliability index (µ)	0.95	0.97
Separation index	4.58	5.36
Mean	1.72	0.00
Standard deviation	1.43	0.44
Outfit		
Mean Square	0.97	0.97
z-Standard	−0.4	0.0

**Table 3 ijerph-18-04949-t003:** Standardized residual variance using Principal Component Analysis.

Standardized Residual Variance (in Eigenvalue Units)	Empirical (%)
Total raw variance in observations	100.0
Raw variance explained by measures	56.4
Raw variance explained by persons	36.6
Raw variance explained by items	19.8
Raw unexplained variance (total)	43.6
Unexplained variance in 1st contrast	5.2

**Table 4 ijerph-18-04949-t004:** Calibration scaling analysis.

Category Label	Observed Count	Observed (%)	Observed Average	Sample Expect	Infit	Outfit	Structure Calibration	Category Measure
					MNSQ	MNSQ		
1	53	5	−2.90	−3.28	1.42	1.53	none	(−4.81)
2	126	37	2.21	−2.19	1.16	1.30	−3.55	−2.70
3	147	43	−1.09	1.01	1.12	1.10	−1.72	−0.04
4	18	5	−0.03	0.17	1.21	1.10	1.76	2.159
5	1	0	−1.14	0.91	1.88	1.00	3.51	(4.67)

Note: MNSQ, mean square.

**Table 5 ijerph-18-04949-t005:** Index level for the initial model.

Name of Category	Name of Index	Level of Acceptance	Index Level
Absolute fit	χ2/df	>0.05	Significant
	RMSEA	<0.08	0.11
Incremental fit	CFI	>0.90	0.89
	TLI	>0.90	0.87
	IFI	>0.90	0.89
Parsimonious fit	Chi-sq/df	<5.0	697.28

Note: CFI = Comparative fit index; IFI = Incremental fit index; RMSEA = Root mean square error of approximation; TLI = Tucker-Lewis index; χ2/df = Chi-squared/degree of freedom.

**Table 6 ijerph-18-04949-t006:** Index level for the revised model.

Name of Category	Name of Index	Level of Acceptance	Index Level
Absolute fit	χ2/df	>0.05	significant
	RMSEA	<0.08	0.10
Incremental fit	CFI	>0.90	0.94
	TLI	>0.90	0.93
	IFI	>0.90	0.94
Parsimonious fit	Chi-sq/df	<5.0	4.73

Note: CFI = Comparative fit index; IFI = Incremental fit index; RMSEA = Root mean square error of approximation; TLI = Tucker-Lewis index; χ2/df = Chi-squared/degree of freedom.

**Table 7 ijerph-18-04949-t007:** Estimate regression of the constructs and items.

Constructs and items			Estimate
Q28Physiological	<-->	Physiological	0.598
Q29Physiological	<-->	Physiological	0.801
Q30Physiological	<-->	Physiological	0.832
Q31Physiological	<-->	Physiological	0.850
Q32Physiological	<-->	Physiological	0.853
Q33Physiological	<-->	Physiological	0.856
Q34Physiological	<-->	Physiological	0.793
Q35Physiological	<-->	Physiological	0.896
Q36Physiological	<-->	Physiological	0.882
Q37Physiological	<-->	Physiological	0.782
Q38Physiological	<-->	Physiological	0.730
VT_Qemo	<-->	Psychological	0.939
VT_Qbeh	<-->	Psychological	0.908
VT_Qcog	<-->	Psychological	0.929
VT_Qbef	<-->	Psychological	0.914

**Table 8 ijerph-18-04949-t008:** Estimate correlations of the two constructs of the measurement model.

Constructs			Estimate
Physiological	<-->	Psychological	0.828
e3	<-->	e4	0.384
e10	<-->	e11	0.441
e7	<-->	e10	0.531
e1	<-->	e2	0.286
e7	<-->	e11	0.329
e9	<-->	e10	0.206
e4	<-->	e5	0.237
e6	<-->	e7	0.181

## Data Availability

The authors are happy to share anonymized data related to this paper upon receiving a specific request, along with the purpose of that request. Interested parties may contact hayatikk@usm.my.

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
