# Peer review of "Vicarious Traumatization Questionnaire: Psychometric Properties Using Rasch Model and Structural Equation Modeling"

_ijerph, 2021, doi:10.3390/ijerph18094949_

Round 1
Reviewer 1 Report
Dear Authors,
Thank you for your work in this important area of need. I have a few suggestions that would make the work more relevant to wider audiences and applicable to informing interventions:
Background: This could be expanded to provide the reader with a context for the tool and the outcomes that it can inform. The use of the tool, is it valid, evidence based and what can it determine? Who were the different professions that were identified and what were the differences between the professional groups?
Aim: Was the aim to test the tool ? Or were you hoping to learn something about the cohort and their needs related to vicarious trauma? This could be more clearly articulated. Did the participants know that this was the aim? Is there another paper to come with the actual results?
Discussion: I was hoping to see some discussion about what were the identified phycological issues and physical issues that were related to exposure to COVID-19 at work.
Conclusions: Determine if your aims were achieved. Conclusion requires more evidence, and relate to existing evidence.
Author Response
Point 1: Background: This could be expanded to provide the reader with a context for the tool and the outcomes that it can inform. The use of the tool, is it valid, evidence based and what can it determine? Who were the different professions that were identified and what were the differences between the professional groups?
Response 1: -We have edited the text and remove the term “professionals”. We have used the term “healthcare providers” to make it more specific.
(See page 2, line 44).
- We have also rephrased the meaning of vicarious traumatization. We have also added text on the development of the original Chinese version of the questionnaire.
(see page 2 & 3, from line 55 to line 73).
-“Vicarious traumatization refers to harmful changes to oneself due to exposure to others’ traumatic events [6]. In this study, the research tool is based on the validated original Chinese version of the vicarious traumatization questionnaire. The original version was developed from qualitative interviews and adapted international trauma-related scales [7].”
- The reliability of the tool was described in the Research tools of Method section. (see page 3, line 33), and in section 3.1 Rasch model analysis as seen in page 5, line 174).
“In this study, the Vicarious Traumatization Questionnaire underwent a translational process outlined by the Translation and Cultural Adaptation-Principles of Good Practice [9] to ensure the content and face validity. The Vicarious Traumatization Questionnaire consists of 38 items, which are composed of two constructs: physiological (11 items) and psychological (27 items) construct. The psychological construct is subdivided into four subconstructs. Namely, emotional (nine items), behavioural (seven items), cognitive (five items), and life belief (six items) subconstruct [7]. Each question score ranged from 0 (never) to 5 (always). Total raw scores are used. The score ranges from 0 to 190, with higher scores indicating more vicarious traumatization. Cronbach’s alpha for the questionnaire was 0.93, and the values for each dimension ranged from 0.73 to 0.92.” ( see page 3, line85-94).
Point 2: Aim: Was the aim to test the tool? Or were you hoping to learn something about the cohort and their needs related to vicarious trauma? This could be more clearly articulated. Did the participants know that this was the aim? Is there another paper to come with the actual results?
Response 2: - We have rephrased the aim of the study.
(See page 2 &3, line 16, 107-109)
“This study aimed to establish the psychometric properties of the Malay version of the Vicarious Traumatization Questionnaire from its original Chinese version among healthcare providers in response to the COVID-19 pandemic in Kelantan.”
This paper is meant only for assessing the psychometric properties of the scale. There is another paper with the actual results on vicarious traumatization levels, which is currently under review.
Point 3: Discussion: I was hoping to see some discussion about what were the identified phycological issues and physical issues that were related to exposure to COVID-19 at work.
Response 3: There is another paper with the actual results on the levels of vicarious traumatization, the physiological and psychological issues which is currently under review.
Point 4: Conclusions: Determine if your aims were achieved. Conclusion requires more evidence and relates to existing evidence.
Response 4: The conclusion was rephrased.
“The Malay version of the Vicarious Traumatization Questionnaire tested among healthcare providers is valid and reliable after testing for the person, and item fit statistics and polarity and confirming the construct validity.”
(see page 11, line 352-354).
Reviewer 2 Report
The article is not up to a publication in a scientific journal.
To start with, very key starting points are not discussed in the article: What means alternative, alternative to what? What is wrong with the original version (a reference to which is not given), so why do we need a separate Malay version. In general there is no discussion on the solutions to measure vicarious trauma in general, and no literature review on the issue.
Having alternative instruments in general sounds like a bad idea. In general, the research gap is not discussed at all.
If the key contribution is the development of a measurement instrument, that if anything should be presented in the article. This is now not done.
The article hides behind the complex qualitative analysis that is also underlined in the title. Usually the methods used are not manifested in the article title. All the fine statistical analysis goes to spill, as the authors do not explain what they are looking for in the analysis and why.
Sampling methods should be discussed in more detail, especially when convenience sampling is performed.
Author Response
Point 1: The article is not up to publication in a scientific journal.
To start with, very key starting points are not discussed in the article: What means alternative, an alternative to what? What is wrong with the original version (a reference to which is not given), so why do we need a separate Malay version. In general, there is no discussion on the solutions to measure vicarious trauma in general, and no literature review on the issue.
Having alternative instruments in general sounds like a bad idea. In general, the research gap is not discussed at all.
Response 1: The “Disaster Relief Alternative Trauma Scale” was named for the tool in its original Chinese version in the scoring leaflet. However, the original authors have just used the term “vicarious traumatization questionnaire” in its publication (Li, Z.; Ge, J.; Yang, M.; Feng, J.; et al. Vicarious traumatization in the general public, members, and non-members of medical teams aiding in COVID-19 control. Brain, Behavior, and Immunity 2020, doi:10.1016/j.bbi.2020.03.007).
Therefore, to avoid confusion, we have changed the “Disaster Relief Alternative Trauma Scale” to “Vicarious Traumatization Questionnaire”. It is the exact term applied by the original authors in their publication.
(See page 1, line 2).
Point 2: If the key contribution is the development of a measurement instrument, that if anything should be presented in the article. This is now not done.
Response 2: This paper is about testing the psychometric properties of the Malay version questionnaire. We have rephrased the objective and Research tool in the Method section as below:
“This study aimed to establish the psychometric properties of the Malay version of the Vicarious Traumatization Questionnaire from its original Chinese version among healthcare providers in response to the COVID-19 pandemic in Kelantan.” (See page 3, line 69-73).
“In this study, the Vicarious Traumatization Questionnaire underwent a translational process outlined by the Translation and Cultural Adaptation-Principles of Good Practice [9] to ensure the content and face validity. The Vicarious Traumatization Questionnaire consists of 38 items, which are composed of two constructs: physiological (11 items) and psychological (27 items) construct. The psychological construct is subdivided into four subconstructs. Namely, emotional (nine items), behavioural (seven items), cognitive (five items), and life belief (six items) subconstruct [7]. Each question score ranged from 0 (never) to 5 (always). Total raw scores are used. The score ranges from 0 to 190, with higher scores indicating more vicarious traumatization. Cronbach’s alpha for the questionnaire was 0.93, and the values for each dimension ranged from 0.73 to 0.92.” ( See page 3, line 85-94).
Point 3: The article hides behind the complex qualitative analysis that is also underlined in the title. Usually, the methods used are not manifested in the article title. All the fine statistical analysis goes to spill, as the authors do not explain what they are looking for in the analysis and why.
Response 3: Revising the title and the aim of this article, it is now clear that this article only includes the quantitative analysis for the psychometric analysis (item analysis) to test the validity and reliability of the questionnaire.
(See page 1, line 2-3)
(See page 2 &3, line 16, 107-109)
Point 4: Sampling methods should be discussed in more detail, especially when convenience sampling is performed.
Response 4: The text was rephrased as below. Convenient sampling was chosen because it was the most feasible method of sampling during the COVID-19 pandemic.
“Convenient sampling of healthcare providers was applied. Those who responded to the survey in the WhatsApp application and fulfilled the eligibility criteria were included.”
(See page 3, line 80-82)
Round 2
Reviewer 1 Report
Thank you for your revisions, this has helped in some cases. However I feel there are many areas that have not been addressed. More extensive revision is required.
I provided more feedback in my first review. I don’t feel as though this was addressed. I think it is probably because the authors are slicing this work to produce more than one paper. This paper would benefit from more detail relating to the study, which I suspect is being held over for another paper. In which case I suggest that only one paper be published as this one is lacking the context to make an impact.
Author Response
Response from authors: Thank you for reviewing the article. I might have inadequately explained the differences between the two papers. I want to elaborate and emphasize here that these two papers were of different concepts, designs, consisted of independent samples and, therefore, different approaches for the conduct of the study. It was, therefore, impossible to combine these two papers.
Specifically, the current paper is meant only for assessing the psychometric properties of the scale. It was a cross-sectional study. The number of samples was based on definitive or high stakes at 99% confidence with best to poor targeting sample size, suitable for determining the psychometric properties of the scale.
Whereas the other paper is of comparative cross-sectional design and compares the levels of vicarious traumatization between frontline and non-front line healthcare providers. It was conducted after the psychometric properties of the Vicarious Traumatization Questionnaire has been established. For that paper, the number of samples calculated based on comparing two means was 160 non-frontline and 160 frontline healthcare providers.
We were aware that validation and main studies were to be conducted separately. To extend the work of a validation study into the context of applying the outcomes in a single paper is inappropriate. It is because the reliability and validity of items, domains, and the overall scale might change after the scale was tested. Therefore, conveying the scale outcomes based on the initial assessment of the subjects on which the validity and reliability of the scale yet to be tested and determined was deemed unreliable.

Reviewer 2 Report
The authors have taken my comments into account and the paper has now improved to a level that it can be published.
Author Response
Thank you very much for reviewing the article. All your kind suggestions were valuable to improve the article to reach the desired level for publication.